# Impact of Artificial Intelligence on COVID-19 Pandemic: A Survey of Image Processing, Tracking of Disease, Prediction of Outcomes, and Computational Medicine

Khaled H. Almotairi [1], Ahmad MohdAziz Hussein [2,*], Laith Abualigah [3,4,5,6,7,8], Sohaib K. M. Abujayyab [9], Emad Hamdi Mahmoud [10], Bassam Omar Ghanem [11] and Amir H. Gandomi [12,13,*]

1   Computer Engineering Department, Computer and Information Systems College, Umm Al-Qura University, Makkah 21955, Saudi Arabia
2   Deanship of E-Learning and Distance Education, Umm Al-Qura University, Makkah 21955, Saudi Arabia
3   Computer Science Department, Prince Hussein Bin Abdullah College for Information Technology, Al Al-Bayt University, P.O. BOX 130040, Mafraq 25113, Jordan
4   Hourani Center for Applied Scientific Research, Al-Ahliyya Amman University, Amman 19328, Jordan
5   Faculty of Information Technology, Middle East University, Amman 11831, Jordan
6   Applied Science Research Center, Applied Science Private University, Amman 11931, Jordan
7   School of Computer Sciences, Universiti Sains Malaysia, Pulau Pinang 11800, Malaysia
8   Center for Engineering Application & Technology Solutions, Ho Chi Minh City Open University, Ho Chi Minh 700000, Vietnam
9   International College of Engineering and Management, Muscat 112, Oman
10  Department of Internal Medicine, Riyadh Care Hospital, Riyadh 14214, Saudi Arabia
11  School of Educational and Psychological Sciences, Amman Arab University, Amman 11953, Jordan
12  Faculty of Engineering and IT, University of Technology Sydney, Ultimo, NSW 2007, Australia
13  University Research and Innovation Center (EKIK), Óbuda University, 1034 Budapest, Hungary
*   Correspondence: amihussein@uqu.edu.sa (A.M.H.); gandomi@uts.edu.au (A.H.G.);
    Tel.: +966-591-082-327 (A.M.H.)

**Abstract:** Integrating machine learning technologies into artificial intelligence (AI) is at the forefront of the scientific and technological tools employed to combat the COVID-19 pandemic. This study assesses different uses and deployments of modern technology for combating the COVID-19 pandemic at various levels, such as image processing, tracking of disease, prediction of outcomes, and computational medicine. The results prove that computerized tomography (CT) scans help to diagnose patients infected by COVID-19. This includes two-sided, multilobar ground glass opacification (GGO) by a posterior distribution or peripheral, primarily in the lower lobes, and fewer recurrences in the intermediate lobe. An extensive search of modern technology databases relating to COVID-19 was undertaken. Subsequently, a review of the extracted information from the database search looked at how technology can be employed to tackle the pandemic. We discussed the technological advancements deployed to alleviate the communicability and effect of the pandemic. Even though there are many types of research on the use of technology in combating COVID-19, the application of technology in combating COVID-19 is still not yet fully explored. In addition, we suggested some open research issues and challenges in deploying AI technology to combat the global pandemic.

**Keywords:** machine learning; deep learning; artificial intelligence; COVID-19; virus; epidemic

## 1. Introduction

The well-known severe acute respiratory syndrome coronavirus 2 (SARS-CoV-2) contamination, dubbed coronavirus disease 2019 (COVID-19), has posed a universal healthcare issue. The pandemic has affected almost 215 nations across the continents, with more than 643,875,406 million confirmed cases of infection, including 6,630,082 deaths, at a rate of 1.59% deaths from all confirmed cases. However, 506,530,275 have recovered from the infection, at 78.6%, as of 9 December 2022 [1].

The domain of science and technology is performing an essential function in creating a cure for the virus. With the pandemic globally still raging due to the evolvement of new variants (i.e., delta variant), there has been a desperate search for ways to curtail its spread and develop a vaccine for the virus [2]. Early response to the disease in China was made by employing artificial intelligence (AI), such as tracking and tracing patients' travel history through facial recognition cameras, delivery of food and medicines using robots [3], disinfection of public buildings using drone technology [4], and dissemination of infoJessrmation to the public to remain indoors [5]. In addition, AI has been employed in the development of new molecules in the fight against COVID-19 [6], as shown in Figure 1, just as scientists are developing new drugs, along with computer experts aiming to detect people suffering from the disease via medical imaging, including CT scans and X-rays [7,8].

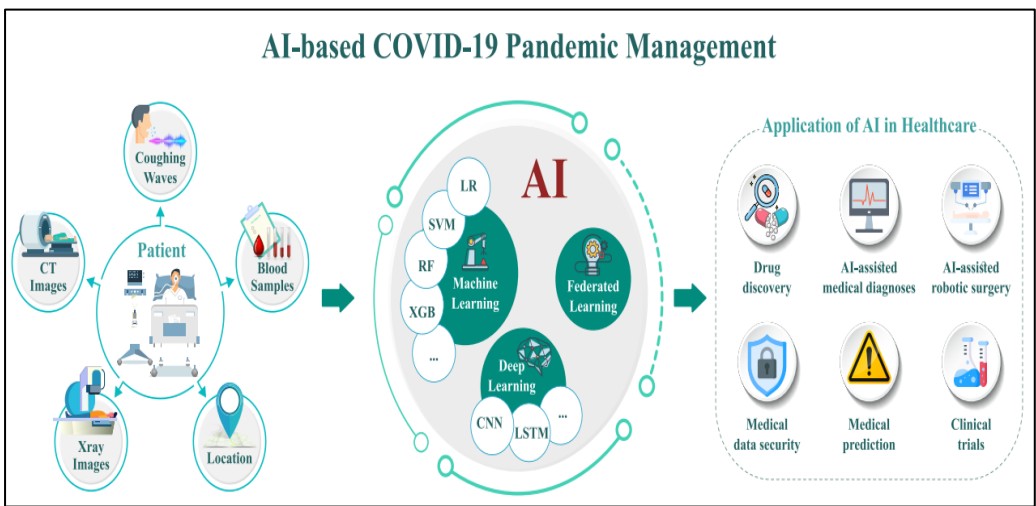

**Figure 1.** AI-based COVID-19 Management Architecture.

Furthermore, with the assistance of AI, tracking innovation is being developed through applications such as monitoring bracelets, which easily track patients breaching lockdown rules. The combination of AI- and mobile phone-aided cameras is also being deployed to take people's body temperature [9]. For example, the national medical insurance database in Taiwan is input with the dataset from both the custom and immigration databases to reconstruct patients' itineraries and symptoms [10,11].

Generally, AI is used to model, forecast epidemics and pandemics, diagnose [12], and validate the healthcare claims of a patient. With the help of supercomputers, different vaccines are being developed for COVID-19 [13]. In addition, drones and robots are deployed for logistics: distributing food and drugs and disinfecting public buildings.

Figure 2 represents the deadliest pandemics and data for the past 102 years. Dengue was discovered in 1950, with about a 100 million–400 million infected persons per year, which leads to about 2.5% of death. Smallpox was discovered and led to the death of about a 300 million people in the 20th century. HIV was discovered in 1920, with more than 75 million infected people and 36 million deaths. Another virus, called rabies, was discovered in 1920, with infection and death rates of 29 million and 5900 yearly, respectively. The Spanish flu was discovered in 1918, which infected more than 500 million every year and led to the deaths of about 50 million–100 million infected people. In 1973, a rotavirus virus was discovered, leading to about 0.2 million–0.5 million deaths yearly.

Afterward, an Ebola virus was discovered, which infected more than 31,000 people and led to the deaths of about 13,000 death tolls. A different virus was discovered, up until the current COVID-19, which was first discovered in 2019, with more than a 200 million infected persons and 4.4 million deaths. Several studies have been conducted to leverage the AI-centred model to enhance the COVID-19 prevention and detection process. In [14], the importance of AI was emphasized for handling the critical stage of COVID-19 prevention

and detection, which is the decision-making stage. Thus, adopting AI would double up and assist in managing patient treatment efficiently in the intensive care unit (ICU).

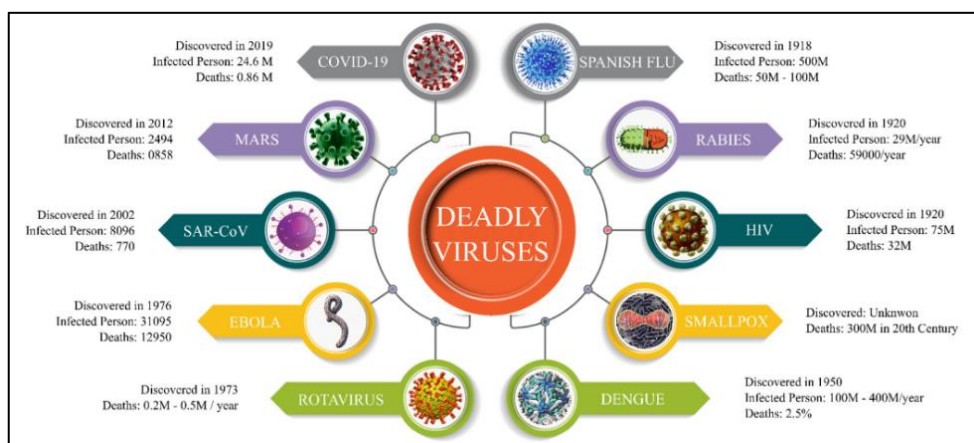

**Figure 2.** Deadliest Pandemics over the last 102 years (as of 25 August 2021).

Naude [10] explored several AI-related research focusing on the COVID-19 pandemic. The use areas of AI for COVID-19 include data dashboards, prognosis cures, diagnosis prediction, and tracking, warning and alert triggering [15], and social control. It is asserted that data scarcity or extensive abundant data employed in data analytics could cause an obstacle for utilizing AI for COVID-19 [16].

*Motivation and Literature Gap*

Several papers have proposed reviews/surveys of applications of AI for curtailing COVID-19. However, the direction and focus vary regarding the characteristics of protocols. For example, in Kumar et al. [17], improved modern technologies for handling the COVID-19 pandemic were reviewed, focusing on the functions of AI and other computer technologies for tackling the pandemic. However, related challenges and the severity of COVID-19 across different countries have not been analyzed. Further, Calandra and Favareto [18] have proposed an overview of the use of AI in combating the COVID-19 outbreak. In addition, dominant variables for AI in combating the COVID-19 outbreak were analyzed. However, current challenges concerning adopting AI for handling the pandemic have not been explored. AI application functions for fighting the spread of COVID-19 have been reviewed [19].

Similarly, a survey on AI and digital style using industry and energy for the post-COVID-19 outbreak has been proposed [20]. However, the research challenges related to security and privacy for adopting AI technologies have not been explored. Hassan et al. [21] proposed a systematic literature review for measuring the impact of AI and mathematical modeling in combating the COVID-19 outbreak. The proposal further surveyed different variants of COVID-19 and quality metrics for evaluating AI and mathematical modeling performances. However, the proposal did not look into the challenges of adopting mathematical modeling and AI paradigms.

In our proposal, we have reviewed different AI technologies and considered their impact on combating the COVID-19 outbreak. An analysis of outbreaks considering different countries is presented. Further, research challenges and open issues focusing on the application of AI for tackling the COVID-19 outbreak have also been proposed. Hence, little or no literature considered the open issues and research challenges in COVID-19 detection and control.

Considering the discussion above, this paper assesses the employment of AI in combating COVID-19. The paper comprehensively reviews the technological advancements at the forefront of the fight against the pandemic. The paper critically examines the AI-based procedures for handling COVID-19. Additionally, this paper advocates the usage of AI. In

addition, the paper explains the deployment of AI and provides context on how innovation is employed against the pandemic. Figure 3 shows the top 17 countries most affected by COVID-19.

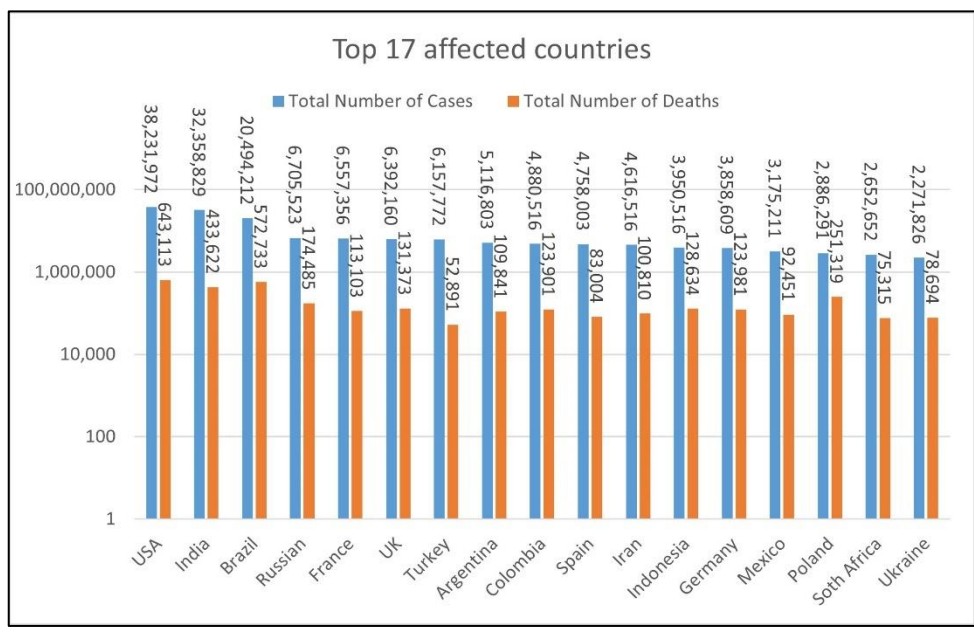

**Figure 3.** Top 17 most affected countries by COVID-19.

The rest of the paper is arranged as follows: Section 2 involves a comparative discussion of related surveys. In Section 3, the analysis of the impact of AI technology on COVID-19 is presented. Further, Section 4 entails a discussion on open research issues and challenges. Lastly, the conclusion and recommendations are presented in Section 5.

## 2. Comparative Discussion of Related Surveys

This section provides a comparative discussion of the related survey, which is further divided into two subsections. Section 2.1 is about the spread of COVID-19, and Section 2.2 involves the diagnostics of COVID-19.

### 2.1. Spread of COVID-19

The ravaging COVID-19 pandemic has changed the direction of research studies because researchers are given more concentration on how to alleviate the virus using various techniques in the AI-centered field. In the interim, researchers have suggested reviews based on AI's function in combating COVID-19 to support relevant authorities, such as medical practitioners [12] and policymakers, in decision-making. The related surveys can be classified into problem-centered AI solutions and AI structures implemented on various COVID-19 processes.

A survey that suggested a classification of tasks involved in predicting the COVID-19 virus has been presented [22,23]. The study outlined the use area of big data and AI. However, most of the considered papers for review are not from reputable sources. In addition, open issues and current research challenges have not been highlighted in the study. In the same direction, Bansal et al. [24] precisely highlighted the function of the AI strategies employed for detecting, predicting, and controlling COVID-19 [25].

Conversely, some COVID-19 processes have not considered some parameters, such as severity assessment and death rates. Further, Kumar et al. [26] concisely extend the function of deep learning (DL) and machine learning networks to handle the pandemic, even though research studies focusing on COVID-19 treatment via respiratory waves and

clinical data have not been explored. Moreover, few studies were explored by analysis of AI-centered applications from different facets [27].

*2.2. Diagnostics in COVID-19*

The foundation of the AI-centered framework and big data concepts used for handling the spread of the COVID-19 pandemic have been reviewed in [28]. Discussions have been provided on the different AI-categorized learning techniques, with specific details on clinical data analysis and results about COVID-19. However, little attention has been given to analyzing the employed techniques. In a similar survey, Swapnarekha et al. [29] classified the reviewed papers into three models, i.e., ML, DL, and statistical, for handling COVID-19 and another related viruses. Further, a summarized review of COVID-19 recognition and prediction is proposed in [30].

A survey based on complicated DL has been proposed by Jamshidi et al. [31]. The survey explored DLs, such as the generative adversarial network (GAN), recurrent neural network (RNN) [32], extreme learning machine, and long short-term memory (LSTM), for a COVID-19 cure. However, the employed models are presented without critical comparative analysis. Further, a description of AI-centered forecasting and statistical model was presented in [33]. On the other hand, the only review of data mining strategies and ML for predicting COVID-19 was proposed in [34,35]. Furthermore, there was a taxonomy for complicated DL techniques for creating radiology reports [36].

Several studies reviewed a certain kind of dataset; for instance, Jalaber et al. [37] put forth the function of CT images for handling COVID-19-infected patients.

The function of the CT scan was also used for handling the presentation of lesions and severity signs. At the end of the paper, five related papers were explored to describe the AI's function for COVID-19 diagnosis. The landscape of radiographic imaging structures and AI methods was investigated. The imaging structures, such as PET, CXR, and CT, were considered for the AI data training and testing. However, the papers considered had constrained information regarding the gained results [38]. In another survey, the imaging characteristics of PET-CT and CT from several articles were presented [39], as well as a comparison of the AI techniques applied for COVID-19 prediction [40]. AI methods for diagnosing COVID-19 have been discussed by categorizing CT and CXR images [41]. In both [42,43], the domain of biosensors and IoT for handling the COVID-19 pandemic have been discussed.

Our paper surveyed articles containing AI's concept for handling COVID-19, in terms of prediction, diagnosis, survival assessment, drug discovery, recasting, and pandemic outbreak. Considering the discussion mentioned above, it is evident that a study focused on a distinct part of COVID-19 handling or described a single type of dataset. Further, many of these reviews offered fewer relative analyses and examined few papers. Conversely, there are a handful of articles that have not been surveyed.

The following section explores and presents the impact of AI in handling COVID-19.

## 3. Impact of AI on Repressing COVID-19

This part discusses the use of artificial intelligence (AI) techniques for handling the COVID-19 pandemic that have been discussed. AI technologies could be based on natural language processing (NLP), ML, and other applications of computer visualization. The different capabilities allow machines to use large information-based frameworks to build, show, and foretell. Table 1 presents numerous uses of technology in the fight against COVID-19. AI is often used to diagnose viruses, analyze medical images, trace, track, and carry out future disease predictions [44]. In addition, it is also used to send alerts to raise awareness and create social awareness virtually.

**Table 1.** Use cases of AI in CT diagnosis for COVID-19 pandemic.

| Country | Authors(s) | AI Technique | Data Size | Correctness Level |
|---|---|---|---|---|
| **China** | [45] | Improved inception transfer-training system | 740 viral pneumonia samples and 325 COVID-19 samples, totaling 1065 CT image data | The sensitivity test result is 0.67 The specificity test result is 0.83 The correctness test result is 79.3% |
| | [46] | Two-dimensional Deep CNN | The sample size for non-positive cases is 1385, with 970 CT capacity, of which 496 patients have been diagnosed with COVID-19 | The specificity test result is 95.47%, The sensitivity test result is 94.06% AUC test result is 97.91% The correctness test result is 94.98% |
| | [47] | It is based on a three-dimensional DL system | An aggregate of 618 CT specimens was gathered of which 219 are from 110 infected persons | The correctness test result is 86.7% |
| | [48] | COVID-19 prediction using neural network called COVNet | An aggregate of 4356 chest CT examinations, which are from 3322 infected persons | The correctness test result is 95% |
| **Canada (Toronto)** | [49] | A deep CNN dubbed COVID-Net: | From 13,645 infected persons, the total of 16,756 CXR images were collected. | The correctness test result is 92.4% |
| **Hong Kong and Thailand** | [50] | The RT-PCR assay is real-time | From 246 infected people, 340 clinical samples were collected | From each reaction or response, more than 10 genomic copies, which is the Potential detection limit |
| **Universal** | [11] | The CXR image from 50 uninfected patients and 50 infected patients with the COVID-19 virus. | Inception ResNet V2 InceptionV3 and ResNet50 | The inception-ResNet at V2 is 87%, the ResNet at 50 is 98%, and the inception at V3 is 97%. |
| **Saudi Arabia** | [51] | COVID-19 detection fuzzy analytic hierarchy process (AHP) | Saudi open data | High efficacy |
| | [52] | Deep learning-based convolutional CNN | Dataset of 340 DX-ray radiographs, 170 images of each Healthy and Positive COVID-19 class. | High precision with maximum accuracy of up to 94.12% |
| | [53] | A dilated CNN and branching design model, and VGG-16 technique | Dataset contains from 13,975 CXR images | Accuracy = 96.5% Sensitivity = 96% |

AI techniques have been used to extract the exact graphical features of COVID-19 that help provide clinical treatment/diagnosis before conducting the pathogenic test, thereby minimizing the time for pandemic control. By employing radiology images in diagnoses, AI obtains radiological characteristics for the prompt and precise discovery of COVID-19 [45]. The techniques employ deep learning algorithms using a computer vision model that considers specific parameters, such as level of specificity, accuracy, sensitivity, region area under the curve (AUC), negative predictive value, and positive predictive value. Similarly, deep convolutional neural networks (CNN), which employ X-ray image data for model training and testing, have been proposed for the automatic detection and prediction of COVID-19 [54]. The proposed technique serves as a substitute treatment and diagnosis decision to avoid the spread of the coronavirus among the infected people around the globe using CNN-based models, which include ResNet50, ResNet101, ResNet152, Inception-ResNetV2, and InceptionV3 [54].

Another solution Wang et al. [49] proposed for handling the pandemic is COVID-Net. It employs the AI concept for detecting coronavirus using the data from an open-source repository of chest X-ray images. Another AI technique has been proposed to screen coronavirus using multiple CNN to classify images and find the probability of the virus infection [47]. The current CT application, and/or the above AI techniques that have been proposed, appear to help ascertain the pandemic to provide diagnosis/clinical support to a patient before conducting the pathogenic result that is ready for proper action.

Wang et al. [55] presented a somewhat effective respiratory simulation model (RSM), in order to handle the limitation between the massive volume of training data and the limited available real data. Meanwhile, the suggested deep learning model could be expanded to big-scale use areas, such as office environments, sleep scenarios, and public places. Although, the technique has faced some challenges, including adequate real-world data to realize the learning method, and the variation in different respiratory patterns is also less than average. The disease tracking procedure involves the following steps: (1) irregular respiratory sequence classifier, which can lead to mass testing of people infected with COVID-19. (2) The SIR model, which is time-bound, is employed for determining the number of infected people. (3) The gated recurrent unit (GRU) is a neural network that uses an embedded bi-directional and attentional system (BI-AT-GRU) for categorizing respiratory sequences. (4) The infectious, exposed, vulnerable, and eliminated or recovered framework is employed to predict the cause of the pandemic.

An ML-based model for predicting the survivability of patients infected with COVID-19 and the prediction result of the patient's state of health has been presented in [56,57]. In [56], the supervised XGBoost classifier gives a straightforward and spontaneous medical screening to measure the likelihood of bereavement accurately and promptly. In [57], the ML-based CT frameworks indicated the possibility and precision of forecasting the stay time of patients infected with COVID-19 at the hospital.

A model is a supporting tool for decision-making and logistical planning for the healthcare system. The technique uses different algorithms with different datasets. Richardson et al. [27] also employed Benevolent AI's knowledge graph to search for approved drugs to help minimize coronavirus infection. The authors did not discuss the detail of the algorithms and how the model performance was evaluated using the available parameters. Similarly, a novel deep-learning pipeline architecture has been proposed [58] as an alternative to COVID-19 detection. The technique uses a chest x-ray image with convolutional CNN to detect whether the patient is a carrier of COVID-19 or not, with detailed diagnosis features and a quicker diagnosis. The technique has been regarded as the most suitable in places that have advanced computing machines. However, during this pandemic, people need a solution that can be integrated with existing and/or available resources. Another technique, based on a cuckoo search optimization algorithm, has been proposed to extract basic information from the X-rays conducted on the lungs using three classification processes: called normal patients, COVID-19-infected patients, and pneumonia patients.

The approach is an alternate solution to detect COVID-19 from the X-ray images using a modified CS algorithm [59,60].

Protein structure prediction is used to extract some features from medical images. In [61], the residual learning procedure was utilized to simplify the training of considerably deep systems for image feature detection. In [62], the critical assessment of methods for protein structure prediction (CASP) by employing a deep neural network to forecast protein characteristics based on its genetic pattern was suggested. In [63], convolutional network architecture was inspected for heavy projection.

Drug innovation is an application for adversarial auto-encoders, which is employed in extracting the method and the structure of image data, dimensionality reduction, unsupervised clustering, and data conception [64]. While in [65], protein structure is used as an incorporated AI-centered drug detection conduit to award new drug mixtures.

In [66], cough-type diagnosis utilized a considerable selection of acoustic characteristics administered to the documented audio from many uninfected and infected persons. In [30], a smartphone thermometer was a simple substitute device for measuring the temperature of infected persons.

Social media has become very popular worldwide, as it is used to interact and communicate [25]. However, one problem is information overload, misinformation, and fake news. To counter this "infodemic", the World Health Organization (WHO) introduced the information network for epidemics (EPI-WIN) to distribute news and data with some major partners [67]. Social media giant Facebook analyzes posts about infections; its ad library [68] examines all ads through the tag "COVID-19" and "coronavirus", and Facebook aggregated 923 outcomes in 34 nations, the maximum of which were from the US (39%) and Europe (Italy had 25% of the ads).

A system for detecting COVID-19 utilizing data from mobile phones' sensors, including cameras, microphones, inertial sensors, and temperature, was proposed in [66]. Similarly, audio data collected from hand-held phones' microphones was employed to identify coughing [30]. It is essential for AI to be trained to predict infection threats and, as such, help identify high-risk cases for containment purposes, thereby curtailing the spread of the virus among the populace [69]. Some drones were also used to trail and detect people who were not using mouth/nose masks, and some were employed as a public address system to address the public or disinfect public places. A company from Shenzhen in China, Small-Multi-Copter, has helped dramatically with logistical support and distribution of medical supplies and lockdown materials via drones.

To curtail the transmission of the virus in India, the authorities introduced Aarogya Setu [70], a mobile phone app that could track coronavirus patients to fight the infection on an individual basis. The app also helped trace contamination using mobile phone GPSs and Bluetooth to collect data on whether a person has come into contact with a COVID-19 patient. To curtail further infection from the coronavirus in India, the authorities developed a mobile application known as Aarogya Setu [70], which tracks coronavirus infection and also aids in stopping the spread from person-to-person. It aids in tracing coronavirus infection by using mobile phones' GPS networks, as well as the Bluetooth of the phones, with which it detects whether an individual has had an interaction with a COVID-19 patient [71].

*3.1. Medical Image Processing*

The effectiveness of the current diagnostics at the beginning of the pandemic was challenged. Open clinical methods were ineffective against the COVID-19 virus, and with limited medical equipment and other assets, the cure needs of every patient were determined by the seriousness of their symptoms. With many outpatients with mild symptoms that could suddenly be serious, there was a need to diagnose the symptoms early enough for effective treatment and ultimately drive down the mortality rate. Therefore, AI could be effective in the prognosis, prediction [72], and curing of COVID-19 patients and drive down treatment costs [73]. Most medical uses of AI are often used for diagnosis using

medical imaging. In some current studies, it was established that only a small number of the studies used AI in arriving at their CT scans. Additionally, other studies employ patients' medical records to predict the severity of the virus [5,45,54].

### 3.1.1. The Role of CT Scan for COVID-19 Patients Screening

The results obtained based on CT from COVID-19 scenarios include multilobar GGO and bilateral with surface or subsequent distribution. This is often in the lower lobes and with less frequency in the intermediate lobe. Subpleural, septal thickening, pleural thickening, and bronchiectasis involvement are a few of the usual outcomes, particularly in the subsequent phase of the virus. CT halo symbol, pleural effusion, lymphadenopathy, pericardial effusion, cavitation, and pneumothorax are also among the few unusual, but probable, outcomes observed from the virus evolution [3,6,7].

Bai and his team noted common features within 201 infected patients, CT irregularities and suitable RT-PCR patients, as follows: 80%, 91%, 56%, and 59% for the surface circulation, GGO, good reticular opacity, and vascular congealing, respectively. Fewer usual features for the CT photographing on the chest included the following: 14%, 2.7%, and 4.1% for the central and peripheral distribution, lymphadenopathy, and pleural effusion, respectively [10]. At an early stage, the chest film is not usually sensitive, and it can be found to be significant at a later stage during the monitoring of the disease [11]. According to Malpani et al. [74], another way of calculating the severity score is to assign the percentages of individuals of the five given lobes shown as <5% contribution, 5–25% contribution, 26–49% contribution, 50–75% contribution, and >75% contribution [8,10]. The overall CT mark contains the summation of each of the given lobar marks that cover the bound of values from 0 to 25 (for no contribution and maximum contribution, respectively), once the contribution of all the five lobes is found to be above 75% [11].

### 3.1.2. Diagnosis Using Radiology Images

With the application of AI, many lives could ultimately be saved, and the spread of the virus could be checked, leading to the generation of relevant data from AI models with correct diagnoses of the virus. With AI, radiologists could achieve faster, not to mention cheaper, diagnosis rates than mainstream coronavirus tests [75]. In the same vein, doctors could also use a combination of X-rays and CT scans [46]. The different AI use cases for handling COVID-19 are presented in Table 1. COVID-19 medical tests are not widely available and often expensive, but most emergency and trauma clinics usually have CT and X-ray machines. Thus, with the help of DL, a radiology expert could analyze and detect the presence of COVID-19. In another development, COVID-Net has proposed an IT-based application for examining COVID-19 signs, based on CXR, through the various information of the lungs of infected patients [76]. Using diagnostic research, AI software was developed from an inceptions migration neural network for analyzing and detecting COVID-19 symptoms with the help of CT images with an 89.5% accuracy rate [45].

A preliminary discovery model has been developed to detect the COVID-19 virus from Influenza-A and specific cases with pulmonary CT images using a DL system. The affected portions of the patients were aggregated using the 3D DL model, and the research had an 86.7% accuracy rate [47,48]. Similarly, Cao et al. [77] built a DL system to effectively diagnose the virus symptoms contracted from other lung ailments and community-acquired pneumonia (CAP). Using chest CT scans, a 3D learning method was developed using a DNN (COV-Net) [78]. Additionally, to diagnose coronavirus, a DL framework was developed that quickly uses CT data as inputs, carries out lung categorization, detects COVID-19, and diagnoses any irregular slice. In addition, the research shows that the diagnoses of AI methods could be explained using data to check the shortcoming of the DNN model as a black box [79]. Subsequently, a computerized system has been developed to quantify the different signs of the virus in patients' lungs and check the virus or response to treatment by employing a DL technique. The range of capabilities of AI clinical analyses have not yet been determined. However, some hospitals in China

have been using AI-aided radiology innovations. Transcription polymerase chain reaction (RT-PCR) tests are critical for diagnosing coronavirus. However, they have their limitations regarding specimen variety and the duration needed for the research and processing [50]. Some abnormalities in CT image data of COVID-19 have been seen using the central AI concept [80,81]. Similarly, the fuzzy-based decision-making technique has been explored by [51] to assess the severity of COVID-19 in the Kingdom of Saudi Arabia (KSA), while adopting a more robust computational model for evaluating the severity using social influence to control the spread of the COVID-19 pandemic [51].

An X-ray technique with an automated system identifier for COVID-19 detection on chest images has been designed using the convoluted CNN architecture. The technique extracts the feature descriptors from the chest X-ray image using a speed-up feature robust algorithm and integrated k-means clustering algorithm to detect whether there is a presence or absence of COVID-19. The study used the dataset of 340 X-ray radiographs and 170 images of both healthy and positive COVID-19 classes [52]. Chest X-ray (CXR) has been used to detect the coronavirus infection by proposing the dilated CNN, branching design model, and VGG-16 technique. Therefore, the VGG-16 used the beginning of the ten layers in the model's front end to extract and utilize the high-level merits [53].

### 3.1.3. Disease Tracking

AI can be deployed for tracing and tracking COVID-19. Current studies have shown that COVID-19 is characterized by respiratory patterns, which differ from normal colds and periodic influenza, including fast breathing (tachypnea) [82]. Predicting tachypnea could become a premium diagnostic characteristic that aids the scope screening of possible patients [55]. So many proposals have been made on how best to employ mobile phones for COVID-19 diagnoses. The best way is either by using embedded sensors that detect COVID-19 symptoms or by conducting phone surveys to help vulnerable patients who depend on responding to critical questions [83].

Berlin uses a model based on epidemiological SIR, which uses curtailment actions by the relevant authorities, such as quarantines, social distance, and partial or total lockdown measures [84]. Another SIR model involves public health methods for handling the virus. It also uses data sources from China and was made available in R [85]. GLEAMviz epidemiological model could be deployed to check the spread of the virus [33]. Similarly, Metabiota [86] uses a tracker for the epidemic to detect COVID-19 [87]. It is also used as a near-tenure forecasting system for the transmission of 93infections. Information about tracking the virus is essential for public health experts to curtail the pandemic [88,89] effectively.

### 3.1.4. Prediction of the Infected Patient

An innovative method that depends on patients' blood tests and medical information was developed to assist doctors in determining vulnerable patients early enough. This will improve virus forecasting and reduce the mortality rate among high-risk patients [56].

Machine learning (ML) has been used extensively to solve various complex challenges in various application areas. ML can help enhance the reliability, performance, predictability, and accuracy of diagnostic systems for many diseases. This paper provides a comprehensive review of the use of ML in the medical field such as detect COVID-19. Such algorithms learn from many diagnosed samples collected from medical test reports. They can also support medical experts in predicting and diagnosing diseases in the future [90].

As an alternative, another forecast model that calculates XGBoost was developed to forecast fatality rates and differentiate the essential factors that can be determined in clinics. The researchers determined three critical factors: high-affectability C-receptive protein, lactic dehydrogenase, and lymphocyte for determining a patient's survivability. The highlight of this approach is its easy convertibility, and the triple factors recognized by the procedure are the important and critical indicators in the pathophysiological progress of COVID-19, especially cell damage, cell inflammation, immunity, and inflammation [91].

A similar study was conducted to predict whether a COVID-19 infected person may need a longer period to stay in the hospital or not, based on a U-Net AI system, which is secondarily trained using CT data [57]. While these methodologies have their shortcomings regarding scope and information, they represent important studies that can be improved with additional clinical information from other cases worldwide. Together, these methods may significantly aid in identifying infected persons needing longer stay periods at the hospital, thereby supporting hospitals in having an adequate plan.

### 3.2. Disease Tracking and Treatment

Computational biologists are essential in combatting the COVID-19 pandemic because of their contributions to modeling. Computational biology is called computational simulation, mathematical modeling, and data analytics for advancing biology [92]. With disease dynamics modeling, the impact of specific parameters that abet disease transmission and mediation's impact in fighting infections is better understood [93]. When a patient dies from the virus, their lungs start manifesting glass and permeating. Different data-aided medication transposition methods were developed to identify diseases, patients, or conditions that could be tackled with the medications used for different ailments [27].

### 3.2.1. Prediction in COVID-19

As soon as a virus RNA penetrates a particular cell, it bonds the affected host cell's protein creation, using it to produce proteins replicating RNA molecules. Proteins possess a 3D structure that can be examined through sequences prearranged by amino acid order. The 3D structure affects the character and objective of the protein [61,92]. They are usually called polymerases and proteins and are the focus of treatments [94].

The two main ways of dealing with forecasts are template modeling, which forecasts structures using the same type of proteins as a framework model succession, and prototype-free modeling, which forecasts patterns for proteins with unidentified associated patterns [62]. It is proposed that these forecasts may assist in finding a cure for the COVID-19 pandemic. Further, the AlphaFold system relies on a bigger ResNet structure and utilizes amino acid order, as well as the characteristics from the corresponding amino acid order through different order structures, to predict the length and the sparsity of gradients among amino acid remains [63]. This method could be used to determine the patterns of the six proteins related to the SARS-CoV-2 layer protein, Nsp2, Nsp4, protein 3a, Nsp6, and proteolytic-like penzyme [61].

### 3.2.2. Discovery of a Drug for COVID-19

At the Massachusetts Institute of Technology (MIT), some experts are currently developing a method for fighting the ravaging COVID-19 by producing a "decoy" receptor or protein, which might be used as a drug. The virus causes illness by attacking and attaching to the body's ACE2 receptors. The experts at MIT are using an AI concept, built on data related to ACE2 receptors, to mimic the link between the hooks and the virus [95]. Few studies are looking at ways to find new composites to focus on SAR-Cov2 by deploying novel conduits to determine constraints for the 3C-similar enzymes [64].

These systems employ three sets of data of the precise architecture of the enzyme, the c-clear substance, and the homogeneity template of the enzyme. Different types of information are used, including the productive automatic-encipher and the productive antipathetic matrix [65]. The researchers are investigating the possibility of utilizing a supplementing cognitive method with a large receptibility that can integrate factors such as the dosages of drugs, similarity, freshness, and different varieties.

## 4. Research Challenges and Open Issues

In this part, we have emphasized some research issues that require research consideration to attain efficient AI technologies for COVID-19 pandemic mitigation. The research issues cut across insufficient data for algorithm training, high computation expenses, secu-

rity and privacy issues, and unclear interoperability functions. The detailed discussion is as follows:

- **Insufficient Data for Machine Training**

  In some parts of the world, there are not sufficient data, such as CXR images from COVID-19-infected persons, for training the machine/algorithm. Similarly, there is no sufficient repository containing all data on the symptoms of infected COVID-19 cases. News and social media data reports may be highly unstructured, multidimensional, and low quality. Data may not be accessible from the community with limited Internet access. In addition, there are challenges in collecting patients' physiological features and therapeutic outcomes. Thus, projecting the skewed outcome of results and erroneous predictions could cause mass hysteria in the healthcare system [16,70]. Considering these challenges, there is a need to build national and global repositories for COVID-19 medical data.

- **High Computational Expenses**

  Since different researchers have mainly employed deep learning (DL) concepts in the quest for combating COVID-19, the machine/algorithm has a high dependency on high-capacity hardware. This is because DL uses a neural network that depends on large datasets for training and testing the COVID-19 prediction model. The need for a large dataset for algorithm training also leads to a long training time, which may not be helpful for the early prediction of the virus [96,97]. Therefore, there is a need to develop more robust DL techniques that consider the urgent need for the COVID-19 predictive model. The model should take less training time in the model training phase.

- **Scarce Data**

  The primary ingredient of machine learning techniques is the large quantity of data. Labeled data are often used to train machine models to learn and make specific predictions. However, with partial data, the whole of the AI system could become flawed. Thus, the large data set might not be readily available in some countries affected by COVID-19. Therefore, there is a need for global repositories where COVID-19 patient data can be accessed.

- **Security and Privacy**

  To restrain the transmission of the virus, mobile applications for the real-time transmission trailing, detection, and observation for quick warning and alerting have been developed. However, the privacy and security of mobile phone users is not explored [76,98]. For instance, using government surveillance gadgets in public places to detect infected persons has sparked adverse reactions from people because such surveillance reveals the identity of every detected individual. Thus, there is a need to ensure that mobile phone data and other surveillance data remain anonymous in the AI technique.

- **Interoperability**

  The data exchange between different nodes or neurons of the learning system is unclear. In DL, interoperability is difficult to detect or understand because it involves complex neurons and operations [99,100]. It is almost unclear how a particular set of inputs leads to a specific solution for various problems. The interoperability issue emanates from either a lack of standardized and coordinated data representation or a standardized application programming interface. In the design of the AI system for COVID-19 prediction, there is a need to design an AI system so that the reasoning behind the operation is evident.

## 5. Conclusions

Scientists are looking at every possible cure for the virus, and modern technology increasingly searches for a possible cure. It is pertinent that technology has become part of our everyday lives; it has additionally now been used in the fight against the coronavirus. This paper highlights the problem of the coronavirus and discusses some algorithms that are practically used in hospitals. The paper also discusses the fact that there is an interest in building a yardstick framework to examine the present methods. The present systems have

precise correctness in predicting COVID-19 symptoms with various types of pneumonia using X-rays scans; however, they do not have both interpretability and transparency. Therefore, we can conclude that technology has many capabilities to overcome the medical and social problems caused by the COVID-19 pandemic. Few such capabilities are advanced and adequate for demonstrating any impact. CT investigation performs a vital function in the mitigation of COVID-19. It was used at the initial detection of the COVID-19 virus, particularly in the extremely vulnerable, asymptomatic occurrences with non-positive PCR tests; CT may perform functions in the following points: triage of patients, estimation of deteriorating, estimation of good cure, and problem handle. The triage of patients can be divided into three categories: possibly with COVID-19, without COVID-19, and seriousness of the infection.

**Author Contributions:** Conceptualization, K.H.A. and A.M.H.; methodology, L.A., A.M.H. and S.K.M.A.; investigation, E.H.M. and B.O.G.; resources, K.H.A. and A.M.H.; writing—original draft preparation, A.M.H. and S.K.M.A.; writing—review and editing, A.M.H.; project administration, K.H.A.; funding acquisition, K.H.A.; writing—review and editing A.H.G. All authors have read and agreed to the published version of the manuscript.

**Funding:** This research was funded by the Deanship of Scientific Research at Umm Al-Qura University (https://uqu.edu.sa) for supporting this work by grant code: (22UQU4320277DS15) to KA. The authors would like to thank the Deanship of Scientific Research at Umm Al-Qura University for supporting this work.

**Institutional Review Board Statement:** Not applicable.

**Informed Consent Statement:** Not applicable.

**Data Availability Statement:** Not applicable.

**Conflicts of Interest:** The authors declare no conflict of interest.

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
