# Peer review of "Impact of Artificial Intelligence on COVID-19 Pandemic: A Survey of Image Processing, Tracking of Disease, Prediction of Outcomes, and Computational Medicine"

_2504-2289, doi:10.3390/bdcc7010011_

Round 1
Reviewer 1 Report
-Paragraph in the introduction is repeated 2 times.
-Please carefully proofread your paper.
-If possible, please expand Section 4.
- Figure 3: y-values in scientific notation, use the y-log-scale
-Figures: increase the resolution
- The journal name is BDCC (now it is Algorithm)
- Refs 63 correct author names
Author Response
REVIEWER 1- COMMENTS
Comment 1
Reviewer #1: Paragraph in the introduction is repeated 2 times.
Response to comment 1
Thank you for your noticed. The duplicated paragraph has been deleted.
Comment 2
Reviewer #1: Please carefully proofread your paper
Response to comment 2
Thank you for the vital comments. The full paper has been revised by native speaker English.
Comment 3
Reviewer #1: If possible, please expand Section 4.
Response to comment 3
Thank you for your observed. Section 4 has been expanded.
Comment 4
Reviewer #1: Figure 3: y-values in scientific notation, use the y-log-scale.
Response to comment 4
Thank you for the necessary remarks. Figure 3 has been updated and we used the y-log-scale.
Comment 5
Reviewer #1: Figures: increase the resolution.
Response to comment 5
Thank you for your comments. The Figures have been increasing the resolution.
Comment 6
Reviewer #1: The journal name is BDCC (now it is Algorithm).
Response to comment 6
Thank you for the needed remarks. The journal name has been updated to BDCC.
Comment 7
Reviewer #1: Refs 63 correct author names.
Response to comment 7
Thank you for the observation. Reference 63 has been written correctly.
Reviewer 2 Report
Dear Editor,
In this paper, the authors present a review englobing the use and impact of artificial intelligence on COVID-19. Although this theme is nowadays of great public interest, this article does not seem to bring innovations, new perspectives, or conclusions that would justify its publication in the current state.
Additionally, there are many other similar articles and reviews about the theme. For example:
• “Applications of artificial intelligence in COVID-19 pandemic: A comprehensive review” (https://pubmed.ncbi.nlm.nih.gov/34400854/)
• “Impact of Artificial Intelligence in COVID-19 Pandemic: A Comprehensive Review” (https://ieeexplore.ieee.org/document/9732091)
• In this link (https://www.bmj.com/AIcovid19), we can find at least eight papers about this theme.
Hence, I suggest the authors reformulate parts of their manuscript, highlighting which points this article is innovative when compared to other publications in the literature.
-------
Minors:
Line 24: “computerized tomography (CT) scan help to diagnose the patient infected by COVID-19.” Change to “computerized tomography (CT) scan helps to diagnose patients infected by COVID-19”.
Line 40: “615,777,70 million”: this number appears to be incorrect.
Line 42: “With the pandemic globally still raging the due evolvement of new variants (i.e., Delta variant) there is a desperate search for ways to curtail its spread and develop”. Change to: “With the pandemic globally still raging due to the evolvement of new variants (i.e., Delta variant), there is a desperate search for ways to curtail its spread and develop”.
Line 56: “health care issue” change to “healthcare issue”
Line 61: “still raging the due evolvement of new variants” => “still raging due to the evolvement of new variants”
Line 94: “diagnosis prediction, and tracking” => “diagnosis prediction and tracking”
line 102: “The Figure 3 shows the top 17 most affected countries by COVID-19.” => “Figure 3 shows the top 17 countries most affect by COVID-19”
Line 110: “Deadliest Pandemics over last 102 years” => “Deadliest Pandemics over the last 102 years”
Line 139: “However, little attention has been given in analysis of the employed techniques.” => “However, little attention has been given to the analysis of the employed techniques”
Line 220: “tool for decision taking” => “tool for decision making”
Line 299: “images with 89.5% accuracy rate” => “images with an 89.5% accuracy rate”
Line 376: what did you mean by “COVID-19 pandtant”?
Line 378: “of disease as well as the impact of mediation” => “of disease, as well as the impact of mediation”
Author Response
REVIEWER 2- COMMENTS
Comment 1
Reviewer #2: Hence, I suggest the authors reformulate parts of their manuscript, highlighting which points this article is innovative when compared to other publications in the literature.
Response to comment 1
We appreciate your comments. we have reviewed different AI technologies and considered their impact on combating the COVID-19 outbreak. An analysis of outbreaks considering different countries is presented. Further, research challenges and open issues focusing on the application of AI for tackling the COVID-19 outbreak have also been proposed. Hence, there are few or no literatures that considered the open issues and research challenges in COVID-19 detection and control. Also, we reformulated some parts in the manuscript and highlighted them in Red colour.
Comment 2
Reviewer #2: Line 24: “computerized tomography (CT) scan help to diagnose the patient infected by COVID-19.” Change to “computerized tomography (CT) scan helps to diagnose patients infected by COVID-19”.
Response to comment 2
Thank you very much for the important comments. The statement has been changed to “computerized tomography (CT) scan helps to diagnose patients infected by COVID-19”.
Comment 3
Reviewer #2: Line 40: “615,777,70 million”: this number appears to be incorrect.
Response to comment 3
Thank you for the observation. The number has been updated till 9 December 2022. Line 42
Comment 4
Reviewer #2: Line 42: “With the pandemic globally still raging the due evolvement of new variants (i.e., Delta variant) there is a desperate search for ways to curtail its spread and develop”. Change to: “With the pandemic globally still raging due to the evolvement of new variants (i.e., Delta variant), there is a desperate search for ways to curtail its spread and develop”. Li
Response to comment 4
Thank you for the necessary remarks. The statement has been changed to “With the pandemic globally still raging due to the evolvement of new variants (i.e., Delta variant), there is a desperate search for ways to curtail its spread and develop”. Line 45
Comment 5
Reviewer #2: Line 56: “health care issue” change to “healthcare issue”.
Response to comment 5
Thank you very much for the important comments. The statement has been changed to “healthcare issue”. Line 41
Comment 6
Reviewer #2: Line 61: “still raging the due evolvement of new variants” => “still raging due to the evolvement of new variants”.
Response to comment 6
Thank you for the necessary remarks. The statement has been changed to “still raging due to the evolvement of new variants”. Line 45
Comment 7
Reviewer #2: Line 94: “diagnosis prediction, and tracking” => “diagnosis prediction and tracking”.
Response to comment 7
Thank you for the needed remarks. The statement has been changed to “diagnosis prediction and tracking”. Line 89
Comment 8
Reviewer #2: line 102: “The Figure 3 shows the top 17 most affected countries by COVID-19.” => “Figure 3 shows the top 17 countries most affect by COVID-19”.
Response to comment 8
Thank you for the comments. The statement has been changed to “Figure 3 shows the top 17 countries most affect by COVID-19”. Line 124
Comment 9
Reviewer #2: Line 110: “Deadliest Pandemics over last 102 years” => “Deadliest Pandemics over the last 102 years”.
Response to comment 9
Thank you very much for the comments. The statement has been changed to “Deadliest Pandemics over the last 102 years”. Line 131
Comment 10
Reviewer #2: Line 139: “However, little attention has been given in analysis of the employed techniques.” => “However, little attention has been given to the analysis of the employed techniques”.
Response to comment 10
Thank you for the comments. The statement has been changed to “However, little attention has been given to the analysis of the employed techniques”. Line 166
Comment 11
Reviewer #2: Line 220: “tool for decision taking” => “tool for decision making.
Response to comment 11
Thank you for the necessary remarks. The statement has been changed to “tool for decision making”. Line 247
Comment 12
Reviewer #2: Line 299: “images with 89.5% accuracy rate” => “images with an 89.5% accuracy rate”.
Response to comment 12
Thank you for the comments. The statement has been changed to “images with an 89.5% accuracy rate”. Line 354
Comment 13
Reviewer #2: Line 376: what did you mean by “COVID-19 pandtant”?.
Response to comment 13
Thank you for the comments. The statement has been changed to “COVID-19 pandemic” Line 431
Comment 14
Reviewer #2: Line 378: “of disease as well as the impact of mediation” => “of disease, as well as the impact.
Response to comment 14
Thank you for the comments. The statement has been changed to “of disease, as well as the impact”. Line 433
Reviewer 3 Report
Overall, this paper reviews a lot of papers related to covid-19, especially AI techniques in medical image processing.
However, the biggest problem of this paper is the organization. In particular, the authors listed and described all related papers in the field without really talking about the connections and comparisons between papers. For example, one possible comparison is the development in covid research over time in the past three years. At the beginning in 2020, the research was mainly focus on small sample within a small geographical area, but later in 2021 and 2022 as people understand covid more, AI techniques improve over time with larger dataset and broader practice areas.
Some detailed comments:
1. The two paragraphs above figure 1 are repeating.
2. Consider use higher resolution images in figure 1 and 2, the words in the two figures are hard to read.
3. The authors need to explain the takeaways from figures, especially figure 2, I don't see why this figure is added here.
4. In section 2, consider split the section into subsections e.g. the first one is the spread of covid and the second one is about diagnostic of covid (e.g. CT images)
5. There is only one table in the paper, but the authors mention table 1 and 2 in the text.
6. The paragraphs above section 3.1 are all talking about covid diagnosis using medical images. This should be put into section 3.1 (3.1.1 and 3.1.2). In section 3, the authors split the section into patient's perspective and biology viewpoint. The paragraphs above section 3.1 should talk about two perspectives in general.
7. In section 3.1, I'm not sure if patient's perspective is a good way to organize the section because everything can relate to patient's perspective. For example, medicine can be related to patient's perspective. One possible split can be 1) medical image processing (3.1.1, 3.1.2 and 3.1.4), disease tracking, treatment (3.2.2), studies about covid (3.2.1).
8. section 3.2.3 talks about misinformation about covid on social media. This should not be under "medicine and computational biology viewpoint". Furthermore, I feel like this paper reviews literature in medical images. Misinformation on social media is a little bit off-topic.
9. In the second paragraph in the introduction, the authors talk about new variants (delta variant). Delta variant is very old. Typo: line 40, 615,777,70 million is not correct
Author Response
REVIEWER 3- COMMENTS
Comment 1
Reviewer #3: The two paragraphs above figure 1 are repeating.
Response to comment 1
Thank you for your noticed. The duplicated paragraph has been deleted.
Comment 2
Reviewer #3: Consider use higher resolution images in figure 1 and 2, the words in the two figures are hard to read.
Response to comment 2
Thank you very much for the important comments. The Figures 1 and 2 have been increasing the resolutions.
Comment 3
Reviewer #3: Consider use higher resolution images in figure 1 and 2, the words in the two figures are hard to read.
Response to comment 3
Thank you for your noticed. The explanation has been added to Section 1.1 Motivation and Literature Gap.
Comment 4
Reviewer #3: In section 2, consider split the section into subsections e.g. the first one is the spread of covid and the second one is about diagnostic of covid (e.g. CT images).
Response to comment 4
Thank you very much for the important comments. Section 2 has been split into two subsections as you mentioned in reviewer point 4.
Comment 5
Reviewer #3: There is only one table in the paper, but the authors mention table 1 and 2 in the text.
Response to comment 5
We appreciate your observation. This is the wrong typing, the correct one is table 1 and the correction has been done.
Comment 6
Reviewer #3: The paragraphs above section 3.1 are all talking about covid diagnosis using medical images. This should be put into section 3.1 (3.1.1 and 3.1.2). In section 3, the authors split the section into patient's perspective and biology viewpoint. The paragraphs above section 3.1 should talk about two perspectives in general.
Response to comment 6
Thank you very much for the important comments. The comments have been correction as the reviewer mentioned.
Comment 7
Reviewer #3: In section 3.1, I'm not sure if patient's perspective is a good way to organize the section because everything can relate to patient's perspective. For example, medicine can be related to patient's perspective. One possible split can be 1) medical image processing (3.1.1, 3.1.2 and 3.1.4), disease tracking, treatment (3.2.2), studies about covid (3.2.1).
Response to comment 7
Thank you very much for the significant explanations. The comments have been corrected as the reviewer mentioned.
Comment 8
Reviewer #3: section 3.2.3 talks about misinformation about covid on social media. This should not be under "medicine and computational biology viewpoint". Furthermore, I feel like this paper reviews literature in medical images. Misinformation on social media is a little bit off-topic.
Response to comment 8
We appreciate your comments. The social media Paragraph has been moved before the section 3.1. Medical Image Processing.
Comment 9
Reviewer #3: In the second paragraph in the introduction, the authors talk about new variants (delta variant). Delta variant is very old. Typo: line 40, 615,777,70 million is not correct.
Response to comment 9
Thank you very much for the significant comments. We reviewed again the WHO platform again (https://covid19.who.int/) and we corrected the numbers. Thanks for noticing this mistake. The correct number is 643,875,406 not 615,777,70 million.
Round 2
Reviewer 1 Report
After reading other reviewer's comments and authors' response, I suggest to accept the manuscript
Author Response
Thank you for your efforts
Reviewer 2 Report
The authors included six new paragraphs to justify the publication. This partly answers some of my main concerns. However, I am still in doubt about the originality of the work.
Indeed, there are many other similar studies in this field. Additionally, as I said earlier, there are other published studies with titles almost identical to this work. Perhaps the authors could change the title for something more specific, for example:
"Impact of Artificial Intelligence on COVID-19 Pandemic: a survey about image processing, tracking of disease, prediction of outcomes, and computational medicine".
In addition, I recommend that you do a thorough grammatical review of the text.
Author Response
REVIEWER 2- COMMENTS
Comment 1
Reviewer #2: Indeed, there are many other similar studies in this field. Additionally, as I said earlier, there are other published studies with titles almost identical to this work. Perhaps the authors could change the title for something more specific, for example:
"Impact of Artificial Intelligence on COVID-19 Pandemic: a survey about image processing, tracking of disease, prediction of outcomes, and computational medicine".
Response to comment 1
Thank you for your noticed. The title has been changed.
Comment 2
Reviewer #2: In addition, I recommend that you do a thorough grammatical review of the text.
Response to comment 2
Thank you for the vital comments. The full paper has been revised grammatically by a native speaker of English.
Reviewer 3 Report
The revision looks good. I just have some presentation comments on references:
Some references are incomplete. For example ref 1,8, 56-59, 65, only a url link is provided with no author and/or organization information. I suggest the authors to check the references and make sure they are consistent.
Author Response
REVIEWER 3- COMMENTS
Comment 1
Reviewer #3: The revision looks good. I just have some presentation comments on references:
Some references are incomplete. For example ref 1,8, 56-59, 65, only a url link is provided with no author and/or organization information. I suggest the authors to check the references and make sure they are consistent.
Response to comment 1
We appreciate your comments. The references have been rechecked again and corrected.
Round 3
Reviewer 2 Report
The authors have satisfactorily addressed most of my concerns.